# Risk Assessment of Passive Smoking Based on Analysis of Hair Nicotine and Cotinine as Exposure Biomarkers by In-Tube Solid-Phase Microextraction Coupled On-Line to LC-MS/MS

**DOI:** 10.3390/molecules26237356

**Published:** 2021-12-03

**Authors:** Hiroyuki Kataoka, Sanae Kaji, Maki Moai

**Affiliations:** School of Pharmacy, Shujitsu University, Nishigawara, Okayama 703-8516, Japan; l6kj5bo6f-tdxu5@docomo.ne.jp (S.K.); s1914121tc@shujitsu.jp (M.M.)

**Keywords:** passive smoking, nicotine, cotinine, in-tube solid-phase microextraction (SPME), liquid chromatography-tandem mass spectrometry (LC–MS/MS), risk assessment

## Abstract

Passive smoking due to environmental tobacco smoke is a serious public health concern because it increases the risk of lung cancer and cardiovascular disease. However, the current status and effect of passive smoking in various lifestyles are not fully understood. In this study, we measured hair nicotine and cotinine levels as exposure biomarkers in non-smokers and assessed the risk from the actual situation of passive smoking in different lifestyle environments. Nicotine and cotinine contents in hair samples of 110 non-smoker subjects were measured by in-tube solid-phase microextraction with on-line coupling to liquid chromatography-tandem mass spectrometry, and self-reported lifestyle questionnaires were completed by the subjects. Nicotine and cotinine were detected at concentrations of 1.38 ng mg^−1^ and 12.8 pg mg^−1^ respectively in the hair of non-smokers, with levels significantly higher in subjects who reported being sensitive to tobacco smoke exposure. These levels were also affected by type of food intake and cooking method. Nicotine and cotinine in hair are useful biomarkers for assessing the effects of passive smoking on long-term exposure to environmental tobacco smoke, and our analytical methods can measure these exposure levels in people who are unaware of passive smoking. The results of this study suggest that the environment and places of tobacco smoke exposure and the lifestyle behaviors therein are important for the health effects of passive smoking.

## 1. Introduction

Tobacco smoking and exposure to environmental tobacco smoke are considered risk factors for cancers, cardiovascular diseases and respiratory disease, and the health effects of tobacco smoke have become a serious social problem [1,2,3]. According to a 2020 World Health Organization (WHO) survey, more than 8 million deaths worldwide are caused by tobacco each year, and more than 7 million of those attributed to direct tobacco use and around 1.2 million to indirect exposure through passive smoking [4]. In addition, 40% of children (under 14 years old), 33% of non-smoking men and 35% of non-smoking women are indirectly exposed to tobacco smoke, of whom 166,000 children, 156,000 men and 281,000 women die each year from passive smoking [5,6]. Furthermore, persons with passive smoking have been reported to have a 1.3-fold higher risk of developing lung cancer than those without passive smoking [7]. In addition, wives of husbands who smoke are reported to have twice the risk of developing lung cancer compared to wives of husbands who do not smoke [8]. Therefore, environmental tobacco smoke, including second-hand smoke such as sidestream smoke from cigarettes and exhaled smoke by smokers, and third-hand smoke from clothing, curtains and indoor wallpaper, may be an important risk factor for lung cancer and tobacco-related diseases in non-smokers [6,9].

Passive smoking not only increases the risk of cancer [9,10,11,12,13,14,15] but has also been reported to be associated with the development of cardiovascular disease [16,17,18,19,20,21,22,23], diabetes [6,24,25], metabolic syndrome [26], psychiatric disorders (depression) [27] and cognitive decline [28]. This is a serious problem associated with a variety of adverse health effects, especially for non-smoking children and pregnant women [29,30]. For example, health hazards caused by passive smoking in children are more severe than in adults, bringing increased risk of growth retardation [31,32] and sudden infant death syndrome (SIDS) [33,34] in fetuses born to mothers exposed to tobacco smoke. In addition, passive smoking in infancy has been linked to obesity, diabetes and metabolic syndrome in adulthood [35,36]. Since passive smoking is unintentional and unavoidable in the presence of smokers in the close relatives, there is an urgent need for effective measures to reduce passive smoking worldwide from both public health and clinical aspects [29]. Therefore, it is essential to determine the facts of biological exposure to prevent health hazards caused by passive smoking [1,3,37], however, the current status and effects of passive smoking in various lifestyle behaviors in different environments are not fully understood.

Tobacco-related compounds such as nicotine, its metabolite cotinine and tobacco-specific nitrosamines have been used as biomarkers of exposure to tobacco smoke [6]. However, their half-lives in urine and blood range from a few hours to 3–4 days, and these compounds are rapidly eliminated after exposure to tobacco smoke [6]. In addition, their concentrations in these matrices are much lower in passive smokers than in active smokers, making them unsuitable for assessing the effects of long-term exposure to environmental tobacco smoke in non-smokers. In contrast, hair samples have been frequently used to assess and monitor the bioaccumulation due to long-term exposure to drugs, environmental pollutants, etc. [38,39,40,41], as the amounts of these compounds in the hair are less affected by daily exposure and variations in metabolism than those in other biological matrices [6,42]. Furthermore, hair samples can be collected easily and less invasively and can be stored at room temperature for up to five years [6,43]. Recently, we developed a simple and sensitive method for the determination of nicotine and cotinine [44,45] and tobacco-specific nitrosamines [46] in hair by in-tube solid-phase microextraction (SPME) with on-line coupling to liquid chromatography-tandem mass spectrometry (LC-MS/MS) [47,48,49] and assessed differences in levels of these compounds in active and passive smokers. In the present study, we measured hair nicotine and cotinine levels in healthy non-smokers and asked self-reported lifestyle questions to assess the risk from the actual status of passive smoking in different lifestyle environments.

## 2. Results

The automated on-line in-tube SPME/LC−MS/MS method [45] is sufficiently selective, sensitive and precise for nicotine and cotinine analysis (Appendix A). This method was successfully applied to the analysis of nicotine and cotinine at pg levels per 1 mg of hair samples without any interference peaks. Typical chromatograms obtained from 200 pg mL^−1^ standard solution and a sample corresponding to 0.2 mg of hair were shown in Figure 1. In this study, the analysis method is the same as in the previous paper [45], but the number of non-smokers was increased from 58 to 110, and a questionnaire on lifestyle was administered to analyze the relationship with the exposure level. In a previous study [45], the levels of nicotine and cotinine in the hair of eight smokers were 43.12 ng mg^−1^ and 655 pg mg^−1^, respectively, and those of 20 non-smokers (not even passive smokers) were 1.86 ng/mg and 8 pg mg^−1^, respectively. The levels of nicotine and cotinine were 23 and 82 times higher in smokers than in non-smokers, respectively. In the present study, the contents of nicotine and cotinine in the hair of 110 non-smokers differ by more than 100-fold in concentration, and were 1.38 ± 1.36 ng mg^−1^ and 12.8 ± 13.7 pg mg^−1^, respectively. In particular, the levels of hair nicotine and cotinine in 46 non-smokers (not even passive smokers) were 1.11 ng mg^−1^ and 8 pg mg^−1^, respectively, similar to the previous results. As shown in Table 1 (Question 1 and 2), however, there was no significant difference in these contents by sex and age of participants.

The nicotine and cotinine contents obtained from hair analyses of 110 non-smokers were compared with data of reported lifestyle factors, such as daily sleeping time, stress, passive smoking, food consumption frequency and type. As shown in Table 1, the levels of nicotine or cotinine in hair were not affected by sleeping time or degree of stress. In this table, “Sometimes” means three days and under per week and “Frequently” means four days and over per week. In the questionnaire survey of passive smoking frequency among 110 non-smokers, 46 reported never, 55 sometimes and 9 frequent or always. Their hair nicotine contents were 1.11 ± 1.11, 1.52 ± 1.54 and 1.94 ± 1.18 ng mg^−1^, and hair cotinine contents were 8.1 ± 4.2, 13.8 ± 11.8 and 31.0 ± 31.9 pg mg^−1^, respectively. Both compounds were detected in the hair of non-smokers who reported never being exposed to environmental tobacco smoke, but at significantly lower concentrations than in those who reported exposure. Whisker-box plots displaying the medians and interquartile ranges of hair nicotine and cotinine are presented in Figure 2A,B. These results indicate that the greater the awareness of passive smoking, the greater the accumulation of both compounds in the hair. Furthermore, the levels of nicotine and cotinine in hair were not affected by the presence or absence of exposure to non-cigarette smoke, such as cooking or wood burning, indicating that passive smoking could be detected selectively by hair analysis.

There were no significant differences in hair nicotine and cotinine levels between frequencies of food intake, but those who consumed more spices tended to have lower levels. On the other hand, there were no significant differences in hair nicotine and cotinine levels depending on the frequency of meat consumption, but these levels tended to increase in those who ate grilled meat more often. Hair nicotine contents by consumption of boiled (9), fried (28) and grilled (73) meat were 0.81 ± 0.59, 1.02 ± 0.80 and 1.59 ± 1.55 ng mg^−1^, and hair cotinine contents were 8.6 ± 2.7, 9.6 ± 6.3 and 19.1 ± 20.5 pg mg^−1^, respectively. Whisker-box plots of medians and interquartile ranges of hair nicotine and cotinine are shown in Figure 3A,B. These results may be affected by the tobacco smoke environment in places where grilled meat is eaten, such as grilled meat restaurants, since there was almost no effect of nicotine and cotinine from smoke other than tobacco smoke, such as smoke generated by cooking.

## 3. Discussion

Using the on-line in-tube SPME/LC-MS/MS method, the effects of tobacco smoke exposure can be assessed selectively and sensitively by analyzing as little as 1 mg of hair. Concentrations of nicotine and cotinine in hair reflect long-term exposure to tobacco smoke and are useful to assess the health effects of environmental tobacco smoke even in people unaware of passive smoking. Although cotinine concentrations in the hair of non-smokers were much lower than nicotine concentrations, there was a marked difference in cotinine concentrations between those who were aware of passive smoking and those who were not. This means that cotinine is more reflective of the reality of passive smoking than nicotine.

The consumption of food affected the accumulation of nicotine and cotinine in the hair. Those who consumed spices more frequently had lower levels of nicotine and cotinine in their hair, suggesting that spices help to remove tobacco compounds from the body. This may be related to the preventive effect of spices such as curcumin on lung cancer [50]. As nicotine and cotinine levels in hair are not affected by cooking smoke, high levels of nicotine and cotinine in the hair of non-smokers who eat grilled meat more frequently suggest that they may be affected by passive smoking in places such as grill restaurants and barbecue. Indeed, in Japan, smoking is relatively common in environments where groups of people drink alcohol and eat grilled meat. People unaware of passive smoking may have been unconsciously exposed to low concentrations of environmental tobacco smoke over long periods of time since nicotine and cotinine were detected in their hair. Therefore, it is important to understand the actual situation of exposure to environmental tobacco smoke because it is impossible to know when and where one will be exposed to passive smoking unless smoking itself is banned, and non-smokers may be forced to smoke passively for long periods of time in a variety of environments. In the future, in order to avoid the risk of passive smoking, it may be possible to prevent health problems caused by passive smoking not only by avoiding exposure to tobacco smoke, but also by improving eating habits and lifestyle habits.

## 4. Materials and Methods

### 4.1. Reagents and Standard Solutions

Nicotine and cotinine were purchased from Sigma-Aldrich Japan (Tokyo, Japan). Nicotine-d_3_ (isotopic purity 98.4%), and cotinine-d_3_ (isotopic purity 99.9%) purchased from Toronto Research Chemicals Inc. (Toronto, Ontario, Canada) were used as an internal standard (IS). Stock solutions of 1 mg mL^−1^ of each compound were prepared by dissolving in methanol. The working solutions were prepared by diluting these stock solutions with distilled water to the required concentration. These prepared solutions were stored at 4 °C in refrigerator until use. Methanol and distilled water were of LC–MS grade, while all other chemicals were of analytical reagent grade.

### 4.2. On-Line Automated Analysis System and Analytical Conditions

Nicotine and cotinine were measured using an on-line automated analysis system comprising in-tube SPME coupled with LC-MS/MS equipped with a Model 1100 series LC (Agilent Technologies, Böblingen, Germany) and an API 4000 triple quadruple mass spectrometer (Applied Biosystems, Foster City, CA, USA) by the previously reported method [45]. LC−MS/MS data were analyzed using Analyst Software 1.3.1 (Applied Biosystems). An outline of the system is shown in Appendix A (refer to Appendix A).

A Carboxen 1006 PLOT capillary column (Carboxen molecularsives, film thickness 17 μm, 60 cm × 0.32 mm i.d., Supelco, Bellefonte, PA, USA) was used as an in-tube SPME device and was connected between the injection needle and the injection loop of the LC autosampler. All controls for extraction, desorption and injection were programmed by the autosampler software (Appendix A).

A Polar-RP 80A column (50 mm × 2.0 mm i.d., particle size 4 μm, Phenomenex; Torrance, CA, USA) was used for LC separation at a column temperature of 30 °C, with 2.5 mM ammonium formate/methanol (25/75, *v*/*v*) as the mobile phase, at a flow rate of 0.2 mL min^−1^ [45]. Electro spray ionization (ESI)–MS/MS of nicotine, cotinine and their stable isotope-labeled compounds was performed in positive ion mode at 4000 V and 600 °C by multiple reaction monitoring (MRM), and quantification and confirmation were performed by MRM of the protonated precursor molecular ions [M+H]^+^ and the related product ions for each compound by the previously reported method [45]. These MRM transitions and optimal MS/MS parameters are shown in Appendix A.

Using this analysis system, calibration curves for nicotine and cotinine were linear in the range of 5–1000 pg mL^−1^ by comparing peak height ratios with each stable isotope-labeled IS, and detection limits (signal to noise ratio of 3) were 0.45 and 0.13 pg mL^−1^, respectively. Validation data of this method are shown in Appendix A.

### 4.3. Preparation and Analysis of Hair Samples and Lifestyle Questionnaires

Hair samples were provided by 110 healthy Japanese non-smoker volunteers (30 men and 80 women, aged 18–68). All volunteers gave informed consent in writing to the experimental protocol approved by the ethics committee of Shujitsu University (approval code 147; 13 October 2017). The collected hair samples were stored in amber glass desiccator at room temperature until use.

Approximately 10 mg of hair was collected from the back of each subject’s head, washed three times with 1 mL dichloromethane by sonication to remove external nicotine and cotinine from the hair surface, and stored in an amber glass desiccator at room temperature until use. About 1–2 mg of dried hair cut into small pieces with scissors was weighed into a 7 mL screw-cap vial, to which 1.0 mL of distilled water and 4 µL of the mixed IS solution containing 200 pg of nicotine-d_3_ and cotinine-d_3_ were added, and the vial was heated and extracted at 80 °C for 30 min with the cap. An aliquot of the extract was transferred to a 2.0-mL autosampler vial and made up to 1 mL with distilled water, and then used for in-tube SPME/LC−MS/MS analysis. The contents of nicotine and cotinine were calculated as ng mg^−1^ and pg mg^−1^ hair, respectively, using calibration curves of each compound.

For all non-smoker subjects, the frequencies of lifestyle behaviors, such as passive smoking, food intake and cooking method were self-reported by questionnaire. Significant differences in the contents of nicotine and cotinine in the hair samples of non-smokers between the scores of these items were analyzed by Student’s *t*-test.

## 5. Conclusions

We measured hair nicotine and cotinine levels using an on-line in-tube SPME/LC-MS/MS method in 110 Japanese non-smokers who completed to a self-reported lifestyle questionnaire and assessed the effects of environment and lifestyle on these levels. Hair nicotine and cotinine levels were significantly higher in people who were highly aware of passive smoking, and the risk of passive smoking was found to be influenced by the type of food intake and the dietary environment. Nicotine and cotinine in hair are useful biomarkers for assessing the effects of passive smoking due to long-term exposure to environmental tobacco smoke, and our method can analyze the actual exposure with or without awareness of passive smoking. The results of this study show that improving eating habits and lifestyle in various environments are important to avoid the risk of passive smoking.

## Figures and Tables

**Figure 1 molecules-26-07356-f001:**
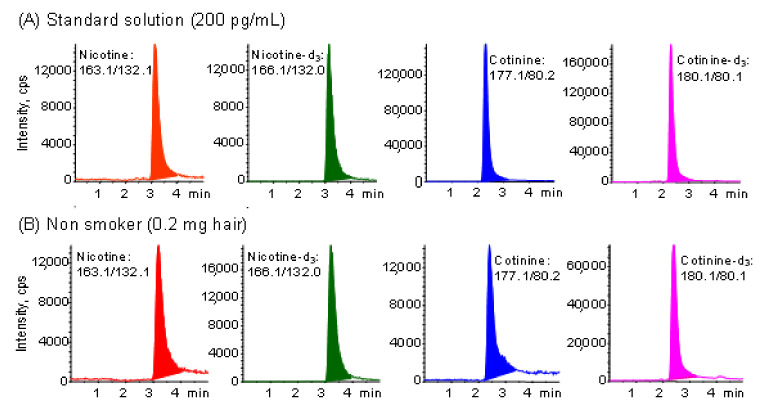
Typical MRM chromatograms obtained from (**A**) standard solution and (**B**) hair sample of non-smoker by in-tube SPME LC−MS/MS. Analytical conditions are described in the Experimental section.

**Figure 2 molecules-26-07356-f002:**
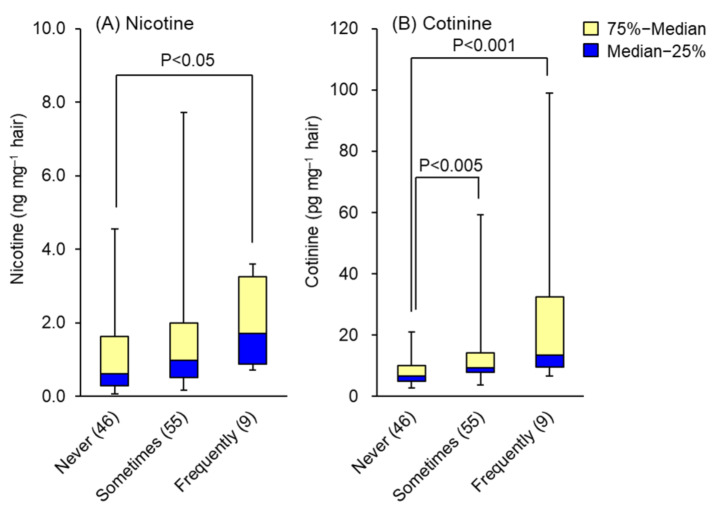
Effects of passive smoking awareness on the levels of (**A**) nicotine and (**B**) cotinine in hair. Data are presented as whisker-box plots displaying medians and interquartile ranges. Not at all: no awareness of passive smoking at all; sometimes: 3 days and under per week; frequently: for 4 days and over per week. The number of subjects is shown in parentheses. Top of each box, 75th percentile; bottom of each box, 25th percentile; solid center line, 50th percentile; error bars, non-outlier range.

**Figure 3 molecules-26-07356-f003:**
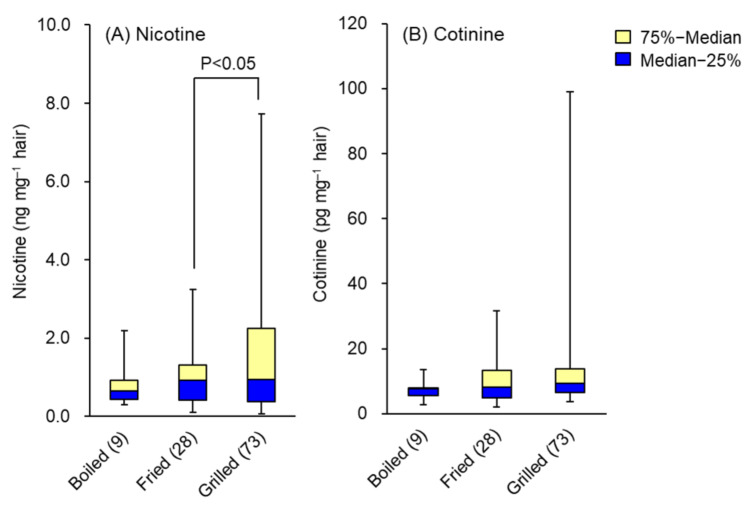
Effects of cooking method of meat on the levels of (**A**) nicotine and (**B**) cotinine in hair. Data are presented as whisker-box plots displaying medians and interquartile ranges. The number of subjects is shown in parentheses. Top of each box, 75th percentile; bottom of each box, 25th percentile; solid center line, 50th percentile; error bars, non-outlier range.

**Table 1 molecules-26-07356-t001:** Nicotine and cotinine contents in non-smoker’s hair based on lifestyle questionnaires.

Question/Answer	*n*	Content in Hair/Mean (Min.~Max.)	
		Nicotine (ng mg^−1^)	Cotinine (pg mg^−1^)
1. Sex			
Male	30	1.63 (0.20~7.72)	14.4 (4.8~43.9)
Female	80	1.29 (0.07~4.99)	12.2 (2.7~99.1)
P ^1^ (Male/female)		0.120	0.234
2. Age			
29 years old and under	79	1.39 (0.07~5.40)	12.1 (2.7~99.1)
30 years old and over	31	1.37 (0.17~7.72)	14.6 (4.3~59.3)
P (Younger/older)		0.477	0.198
3. Daily sleeping time			
Less than 6 h	30	1.19 (0.07~3.89)	10.8 (4.1~59.3)
More than 6 h	80	1.45 (0.10~37.72)	13.6 (2.7~99.1)
P (Shorter/longer)		0.188	0.171
4. Stress awareness			
Sometimes ^2^	68	1.39 (0.07~7.72)	13.8 (2.7~99.1)
Frequently ^2^	42	1.38 (0.17~4.99)	11.2 (4.4~59.3)
P (Sometimes/frequently)		0.483	0.169
5. Passive smoking awareness			
Never	46	1.11 (0.07~4.55)	8.1 (2.7~21.0)
Sometimes	55	1.52 (0.16~7.72)	13.8 (3.8~59.3)
Frequently	9	1.94 (0.72~3.60)	31.0 (6.6~99.1)
P (Never/sometimes)		0.064	0.001
P (Never/frequently)		0.023	0.00001
6. Exposure to other smoke			
Never	55	1.35 (0.12~5.40)	14.0 (2.7~99.1)
Sometimes~Always	55	1.42 (0.07~7.72)	11.6 (3.8~47.0)
P (Never/yes)		0.397	0.177
7. Frequency of tea drinking			
Sometimes/Frequently	46	1.49 (0.17~7.72)	12.4 (4.0~59.3)
Always	64	1.30 (0.07~5.40)	13.1 (2.7~99.1)
P (Less/more)		0.241	0.409
8. Fat and fatty food intake			
Sometimes	8	1.82 (0.37~7.72)	12.7 (5.1~35.2)
Frequently	81	1.27 (0.07~4.99)	11.8 (2.7~99.1)
Always	21	1.64 (0.20~5.40)	16.9 (3.8~47.0)
P (Frequently/always)		0.405	0.227
9. Vegetable intake			
Sometimes	31	1.27 (0.10~4.54)	9.7 (2.7~32.5)
Frequently	64	1.36 (0.07~7.72)	15.1 (4.0~99.1)
Always	15	1.71 (0.20~4.99)	9.4 (3.8~22.0)
P (Sometimes/always)		0.135	0.450
10. Consumption of vegetables			
Raw	55	1.29 (0.07~7.72)	12.0 (2.7~99.1)
Boiled	21	1.30 (0.12~3.60)	13.0 (4.3~59.3)
Pan-fried	34	1.57 (0.10~5.42)	14.0 (3.8~67.0)
P (Raw/pan-fried)		0.232	0.390
P (Boiled/pan-fried)		0.192	0.249
11. Spice use			
Sometimes	36	1.54 (0.12~5.40)	12.7 (4.0~59.3)
Frequently	58	1.41 (0.10~7.72)	13.7 (2.7~99.1)
Always	16	0.93 (0.07~3.48)	9.7 (4.9~31.7)
P (Sometimes/always)		0.055	0.169
12. Meat intake			
Sometimes	14	1.20 (0.28~4.99)	10.5 (5.1~31.7)
Frequently	67	1.47 (0.07~7.72)	13.5 (2.7~99.1)
Always	29	1.28 (0.12~3.60)	12.2 (4.8~67.0)
P (Sometimes/always)		0.414	0.311
13. Consumption of meat			
Boiled	9	0.81 (0.30~2.19)	7.6 (5.1~13.7)
Pan-fried/Deep-fried	28	1.02 (0.10~3.25)	9.6 (2.7~31.7)
Grilled	73	1.59 (0.07~7.72)	19.1 (4.0~99.1)
P (Boiled/grilled)		0.071	0.096
P (Fried/grilled)		0.034	0.053
14. Seafood intake			
Sometimes	48	1.40 (0.12~5.40)	12.0 (2.7~47.0)
Frequently	54	1.37 (0.07~7.72)	14.1 (3.8~99.1)
Always	8	1.24 (0.17~3.44)	8.9 (5.6~1.45)
P (Sometimes/always)		0.478	0.205
15. Consumption of seafood			
Raw	21	1.12 (0.12~4.99)	17.9 (4.4~99.1)
Boiled	33	1.38 (0.29~5.40)	12.6 (4.0~43.9)
Grilled	56	1.48 (0.07~7.72)	11.0 (2.7~67.0)
P (Raw/grilled)		0.155	0.036
P (Boiled/grilled)		0.369	0.228

^1^ Probability (significant difference T-test). ^2^ Sometimes: 3 days and under per week; frequently: 4 days and over per week.

## Data Availability

The data presented in this study are available on request from the corresponding author.

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
