# Peer review of "Risk Assessment of Passive Smoking Based on Analysis of Hair Nicotine and Cotinine as Exposure Biomarkers by In-Tube Solid-Phase Microextraction Coupled On-Line to LC-MS/MS"

_molecules, 2021, doi:10.3390/molecules26237356_

Round 1

Reviewer 1 Report

Risk Assessment of Passive Smoking Based on Analysis of 
Hair Nicotine and Cotinine as Exposure Biomarkers by In-Tube
Solid-Phase Microextraction Coupled On-Line to LC-MS/MS  is presented in this manuscript. The manuscript is well organised and well written and can be recommended for publication.

Author Response

Risk Assessment of Passive Smoking Based on Analysis of Hair Nicotine and Cotinine as Exposure Biomarkers by In-Tube Solid-Phase Microextraction Coupled On-Line to LC-MS/MS is presented in this manuscript. The manuscript is well organised and well written and can be recommended for publication.

We are very grateful to you for the time and efforts entailed in the reviewing process of our manuscript. We are glad to know that our paper can be recommended for publication.

Reviewer 2 Report

The article MOLECULES "Risk assessment of passive smoking..." shows a discussion on the effect of passive smoking related to lifestyle. The article is well organized and the conclusions are interesting. Some aspects should be discussed before accepting the paper:

  1. The novelty of the paper should be better explained because the article seems to be a continuation of reference 45 where the experimental measurements seem to have been already presented.
  2. Nicotine and cotidine levels for smokers and non-smokers (not even passive) should be included as reference values. Perhaps taken from the bibliography.
  3. In Figure 2, the meaning of “Not at all”, “sometimes”, “frequently” should be indicated. If it is the exposure of the passive smoker an indication of the meaning in exposures per week would be clarifying.

Author Response

The article MOLECULES "Risk assessment of passive smoking..." shows a discussion on the effect of passive smoking related to lifestyle. The article is well organized and the conclusions are interesting. Some aspects should be discussed before accepting the paper:

We are very grateful to you for the time and efforts entailed in the reviewing process of our manuscript. Thank you for your valuable comments and advise. As follows, we have tried to revise the paper based on the comments, and have highlighted the responses and changes made in red.

Comments and Suggestions for Authors

The novelty of the paper should be better explained because the article seems to be a continuation of reference 45 where the experimental measurements seem to have been already presented.

Response: Thank you for your appropriate advice. In this study, the analysis method is the same as in the previous paper, but the number of non-smokers was increased from 58 to 110, and a questionnaire on lifestyle was administered to analyze the relationship with the exposure level. These information have been added in the text (lines 85-87).

Nicotine and cotinine levels for smokers and non-smokers (not even passive) should be included as reference values. Perhaps taken from the bibliography.

Response: We thank for your valuable comments. In a previous study [45], the levels of nicotine and cotinine in the hair of 8 smokers were 43.12 ng mg-1 and 655 pg mg-1, respectively, and those of 20 nonsmokers (not even passive smokers) were 1.86 ng mg-1 and 8 pg mg-1, respectively. The levels of nicotine and cotinine were 23 and 82 times higher in smokers than in non-smokers, respectively. In the present study, 46 nonsmokers who were completely unaware of passive smoking had similar results, with nicotine and cotinine levels in their hair of 1.11 ng mg-1 and 8 pg mg-1, respectively. These levels have been added in the text (lines 87-95) as reference values.

In Figure 2, the meaning of “Not at all”, “sometimes”, “frequently” should be indicated. If it is the exposure of the passive smoker an indication of the meaning in exposures per week would be clarifying.

Response: Thank you for your valuable advice. The meanings of "not at all," "sometimes," and "frequently" in Figure 2 are the same as in the footnotes in Table 1 (lines 101-103 in the text) . As you pointed out, they are also shown in the figure captions (lines 166-168) as follows. Not at all: no awareness of passive smoking at all; Sometimes: 3 days and under per week; Frequently: for 4 days and over per week.